# Integrative Taxonomy Supports Two New Species of *Rhodiola* (Crassulaceae) in Xizang, China

**Shiyong Meng \*** , **Zimeng Wang and Lv Ye**

School of Life Sciences, Peking University, Beijing 100871, China; xztswzm@126.com (Z.W.);
yelv_123@163.com (L.Y.)
\* Correspondence: msy542702@pku.edu.cn

**Abstract:** The Qinghai–Tibet Plateau includes the Himalayas and Hengduan Mountains and is well known for its rich biodiversity. Evolutionary radiation is one of the main ways by which plants diversify in mountains, particularly the Qinghai–Tibet Plateau. It presents a large challenge to the classification of taxa that radiate quickly. One way to overcome these challenges is to continue conducting detailed field studies while integrating morphological and molecular evidence to classify these taxa. The aim of this research was to provide a case for the systematic study of the complex taxa *Rhodiola*, which rapidly radiate. During the field study, we found two unique variants of *Rhodiola* in an alpine dry meadow and beds of pebbles on beaches, respectively. We utilized a morphological principal component analysis, scanning electron microscopy and molecular phylogenetic analysis to propose two new species: *Rhodiola wangii* S.Y. Meng and *Rhodiola namlingensis* S.Y. Meng. *R. wangii* is similar to *R. stapfii* (Hamet) S.H. Fu, but it differs in having an intensely broad rhombus and alternate leaves, a distinct petiole, stamens gathered together and reflexed purple scales. *R. namlingensis* is similar to *R. prainii* (Hamet) H. Ohba, but it differs in its exerted alternate leaves, the presence of more than four leaves on the stem, thick leaf blades, an obovate to inverted triangle, and short petioles. The conservation status of these two species was also assessed.

**Keywords:** Xizang; *Rhodiola namlingensis*; *Rhodiola wangii*; new species; taxonomy; radiation

## 1. Introduction

The Qinghai–Tibet Plateau (QTP) and adjacent high-altitude areas contain more than 12,000 species of flowering plants, which renders it one of the areas of the world with the highest richness of species and a high level of endemism because of its habitats, such as high mountainous ranges, steep gorges, rocky outcrops, desert steppes and alpine meadows among others [1,2].

Evolutionary radiation is one of the main ways by which plants diversify in mountains, particularly the Qinghai–Tibet Plateau [3,4]. However, radiated taxa are often inconsistent between molecular phylogenetic and morphological classifications [5,6]. It is possible that these inconsistencies are caused by inadequate sampling or limited genomic sampling with insufficient molecular markers. In addition, some morphological characters are homogeneous [6]. Therefore, it is essential to recruit more molecular markers while conducting more field surveys to find valid characters and more types of variation. Several attempts by different types of molecular markers have been made to define the species of these genera [7–9], but the importance of field study has been neglected. However, hundreds of new plant species in the Qinghai–Tibet Plateau have been described in recent years [10], indicating that the surveys conducted so far are highly insufficient.

*Rhodiola* L. are members of the Crassulaceae DC., which includes more than 70 different species that are primarily distributed throughout the alpine zone of the northern hemisphere, particularly in the Qinghai–Tibet Plateau and adjacent high-altitude areas. Their major characteristics include well-developed thick rhizomes and apical parts with scaly

leaves [11–13]. *Rhodiola* species occupy many habitats. For example, *R. yunnanensis* (Franch.) S.H. Fu, *R. macrocarpa* (Praeg.) S.H. Fu, and *R. liciae* (Hamet) S.H. Fu are distributed in thickets at low-to-medium altitudes; *R. sacra* (Prain ex Hamet) S.H. Fu and *R. bupleuroides* (Wall. ex Hook. f. et Thoms.) S.H. Fu grow in rocky outcrops at medium-to-high altitudes, and *R. coccinea* (Royle) Borissova and *R. crenulata* (Hook. f. et Thoms.) H. Ohba occupy mountain tops that exceed 4500 m. *Rhodiola* are highly diverse morphologically, and Fu and Fu [14] divided the genus into eight sections with seven series based on morphological studies. However, molecular phylogenetic research showed that only three of the eight sections (Sect. *Trifida* Fu, Sect. *Prainia* H. Ohba, and Sect. *Pseudorhodiola* (Diels) H. Ohba) can be considered monophyletic, while two important characters (dioecy and marcescent flowering stems) evolved multiple times within *Rhodiola* [15]. Molecular dating analysis showed that the primary diversification of *Rhodiola* occurred during two periods of the QTP uplifts, at 15–6.5 mya and 5–1.8 mya ago [15]. Rapid diversification, simplicity and the homogeneity of morphological characters lead to taxonomic difficulties in *Rhodiola*, particularly in the delimitation of species [16]. Thus, further intensive studies are still merited with more extensive sampling to clarify the systematic relationships of *Rhodiola*.

Our aim was to explore the diversity of *Rhodiola* in the Qinghai–Tibet Plateau and define species by combining morphological and molecular evidence. An extensive study of the systematic evolutionary biology research of *Rhodiola* in the wild has been conducted since 2010, which has led to the publication of some new *Rhodiola* species [17]. Here, we propose two new species of *Rhodiola* that are distributed on the platform of the Qinghai–Tibet Plateau. One of them is distributed on the shores of the plateau valley, whereas the other grows in the alpine dry meadow. These new species are very small and can easily avoid detection. The discovery of these two new species will help to understand the rapid radiation of *Rhodiola* in the Qinghai–Tibet Plateau.

## 2. Materials and Methods

### 2.1. Materials Collection and Field Investigation

Habitat plays an important role in the diversification of *Rhodiola* [15], and we took great pains to record that habitat during field investigations. Samples of two putative new species were collected in the field, including one from Lhünzê County (one population, 28°06′00″ N, 91°55′53″ E) and another from Namling County (Namling 1: 30°04′91″ N, 89°06′27″ E; Namling 2: 29°55′56″ N, 89°07′01″ E). Their close relatives were also collected from Xizang Province, China. The type specimens and other specimens of *R. prainii* and *R. stapfii* were examined, and specimens of the other related taxa were obtained from herbaria (CVH, K, PE) and examined for comparative research.

### 2.2. Morphological Analysis

Leaf characters play an important role in the definition of *Rhodiola* species [16], while the flowers are morphologically simple. Thus, the morphological analysis is primarily based on the leaf characters. To avoid the influence of artificiality and find key delimitation traits, we used a principal component analysis (PCA) to perform statistical analyses on the traits. We observed and measured the leaf traits and conducted the PCA using Origin 2020 (OriginLab Corporation, Northampton, MA, USA). All the data were quantitative, including the leaf length (L), leaf width (W), length of petiole (P), relative length of the petiole (B = P/L), the distance between the widest part of blade and the base of petiole (H), relative width of the petiole (K = M/W), relative length of the distance between the widest part of blade and base of petiole (D = H/L) and the shape of the blade (S = W/L) (Figure S1).

### 2.3. Scanning Electron Microscopy (SEM) Analysis

Although Gontcharova et al. [18] showed that the seed coats of *Rhodiola* in the Russian Far East vary considerably, they correspond to the features of gross morphology at the species level. Thus, we conducted a scanning electron microscopy (SEM) analysis using a Helios NanoLab G3 UC (Thermo Fisher, Waltham, MA, USA). First, seeds were directly

fixed on the sample shuttle. Secondly, the sample was coated with gold using a vacuum sputter coater. Finally, the sample was transferred to the stage of the sample room for observation. The terminology of seed coat sculpturing and anatomy were those described by Gontcharova et al. [18]. The sizes of seeds were measured using Photoshop CC 2018 (Adobe, San Jose, CA, USA).

*2.4. Phylogenetic Analysis*

DNA barcoding research showed that the nuclear ribosomal internal transcribed spacer (ITS) was the best single-locus barcode, resolving 66% of the *Rhodiola* [19]. Thus, the ITS regions were used as molecular markers. The DNA of the new species was extracted and amplified by PCR as described by Zhang et al. [17]. The ITS sequences of 33 specimens of *Rhodiola* were downloaded from NCBI along with two accessions of *Phedimus*, another genus of the Crassulaceae, as outgroups. The GenBank accession numbers are shown in Table S1.

Phylogenetic analyses were performed using Bayesian inference (BI) and maximum-likelihood (ML) methods in PhyloSuite [20]. The ITS sequences were aligned with MAFFT v. 7 [21] and manually checked in PhyloSuite [20]. The evolutionary models for the ML and BI analyses were determined by ModelFinder [22] using the Akaike Information Criterion (AIC). ML trees with 1000 bootstraps (BS) were produced using an IQ-tree [23]. BS analyses were used to evaluate the support for individual clades with 5000 replicates. A BI analysis was performed using MrBayes [24]. Four chains of the Markov Chain Monte Carlo (MCMC) were run for 2,000,000 generations, sampling one tree every 100 generations, starting with a random tree. The average standard deviation of the split frequencies was used to assess the convergence of two runs. A majority rule (>50%) consensus tree was constructed after removing the burn-in period samples (the first 25% of the sampled trees) and posterior probabilities (PP) to estimate the robustness of the BI trees.

## 3. Results

*3.1. Habitat*

During the field investigation, we found that these two new species and similar ones grow on alpine meadows, but the habitats can be differentiated from each other. The Lhünzê population grows on the soil slopes of alpine meadows that are relatively dry, while *R. stapfii* grows in the moist parts of alpine meadows. The Namling populations primarily grow on the beds of pebbles on beaches or in rock crevices on the shore of the Xiangqu (Jiacuo Zangbo), which is one of the Yarlung Zangbo's tributaries. In contrast, *R. prainii* primarily grows on rocks or trees in the subtropical rain forest on the southern slope of the Himalayas and the slope rocks on the plateau of the Qinghai–Tibet Plateau.

*3.2. Morphological Analysis*

Morphologically, the Lhünzê population is closely related to *R. stapfii*. However, several characters differentiate them. First, rhizomes of the Lhünzê population are glossy, while those of *R. stapfii* are enfolded by the remnants of old shoots. Secondly, flowering stems with some leaves are alternate in the Lhünzê population, whereas all four to six pieces of the leaves are in whorls and on the apical part of the *R. stapfii* rhizomes. Third, the leaves of Lhünzê population have obvious petioles, while those of *R. stapfii* are very short or sessile. Finally, the stamens of the Lhünzê population are gathered together, and the scales are purple and reflexed. In contrast, the stamens of *R. stapfii* do not gather together, and the scales are white. A principal component analysis (PCA) based on the 8 qualitative leaf traits revealed considerable variability among the 44 individuals considered in this study (Figure 1A). The PC1 scores, which accounted for 48.9% of the total variation, showed very high correlation with length of the leaf (L) and petiole, including the relative width of the petiole (K = M/W) and relative length of the petiole (B = P/L). The scores of PC1 were also highly correlated with other traits, such as shape of the blade (S = W/L), relative length of the distance between the widest part of blade and base of petiole (D = H/L), and the length

between the widest part of blade and base of petiole (H). The PC2 scores, which explained 31.8% of the total variation, were highly correlated with the width of the leaf (W), between the widest part of blade and the base of petiole (H) and length of the leaf (L).

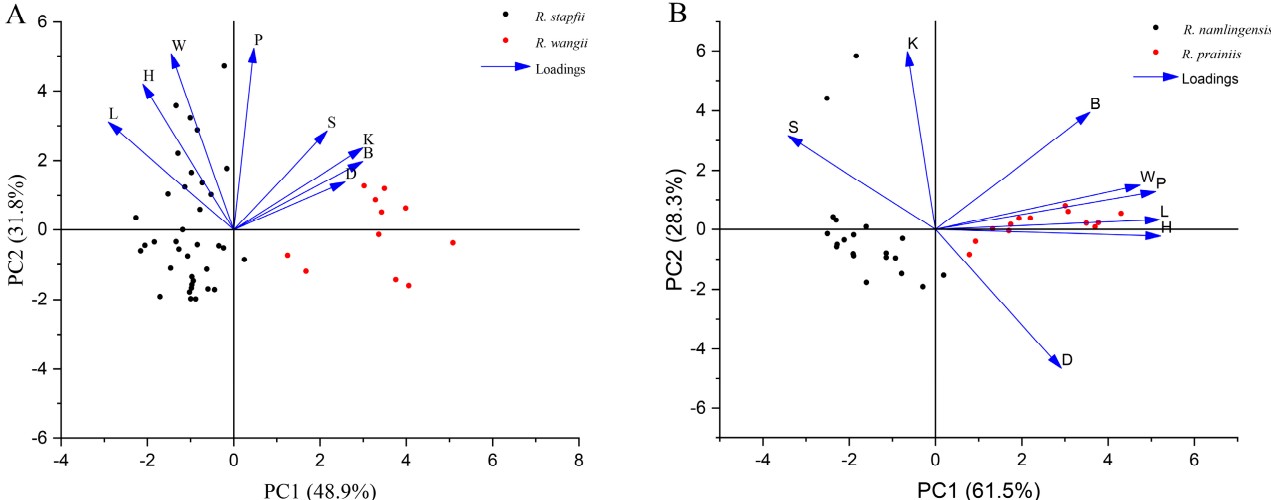

**Figure 1.** Principal component analysis based on the leaf traits. (**A**) Lhünzê population and *Rhodiola stapfii* (Raymond-Hamet) S.H. Fu. (**B**) *R. namlingensis* S.Y. Meng and *R. prainii* (Hamet) H. Ohba.

A morphological analysis indicated that the Namling populations are similar to those of *R. prainii* with flattened leaf blade and similar inflorescences and flower structures. However, the plants can be differentiated by several characters. First, the Namling populations contain more developed flowering stems, while *R. prainii* contains short flowering stems, which usually have only two nodes on the stem. Secondly, leaves on the stems in Namling populations are alternate, while the leaves of *R. prainii* often have four whorls at the top of stem. The lower part of the stem leaves is opposite and has a developed pseudopetiole that is 1 to 3 cm long. Third, *R. prainii* flowers from July to August, while the Namling populations flower from September to mid-October. The PCA based on the eight quantitative traits revealed a considerable variability among the 47 individuals. The first two PCs together explained 89.7% of the total variation (Figure 1B). The PC1 scores, which accounted for 61.5% of the total variation, were showed very high correlation with the length of the leaf (L), width of the leaf (W), length between the widest part of blade and base of the petiole (H) and length of the petiole (P). The scores of PC2, which explained 28.2% of the total variation, were highly correlated with the relative width of the petiole (K = M/W), relative length of the petiole (B = P/L) and relative length of the distance between the widest part of blade and base of petiole (D = H/L).

### 3.3. SEM Analysis

The seeds of *R. stapfii* are triangular with a wing-like projection at the chalazal end, 920 µm long, and 400 µm wide (Figure 2A). The testa cells are morphologically uniform or vary only slightly in their shape and are less regularly arranged. The outer periclinal cell walls are flat and smooth, while the anticlinal cell walls are thickened and bulging (Figure 2E). The cells are quadrangular and isodiametric, and the cell boundaries are well-defined and curved. The outer periclinal cell wall is convex. The seeds of the Lhünzê population are square and lack wing-like projections at the chalazal end, and they are 519 µm long and 324 µm wide (Figure 2B). The testa cells are not readily apparent and have dense stripes in the outer periclinal wall (Figure 2F). The seeds of *R. prainii* are similar to those of *R. stapfii*. They are triangular but small (761 µm long and 309 µm wide) (Figure 2C,G). They lack a wing-like projection at the chalazal end, and the anticlinal cell walls are thin and sunken. The outer periclinal cell wall is thickened and convex. The seeds of Namling populations are oblong and lack a wing-like projection at the chalazal

end, and they are 835 μm long and 203 μm wide (Figure 2D). The testa cells have obvious longitudinal edges on the surface; the width between the longitudinal edges is 23.6 μm, and there are obvious horizontal stripes in the depressions between the edges that extend to the longitudinal edges, forming a regular pattern of orbital decoration. (Figure 2H).

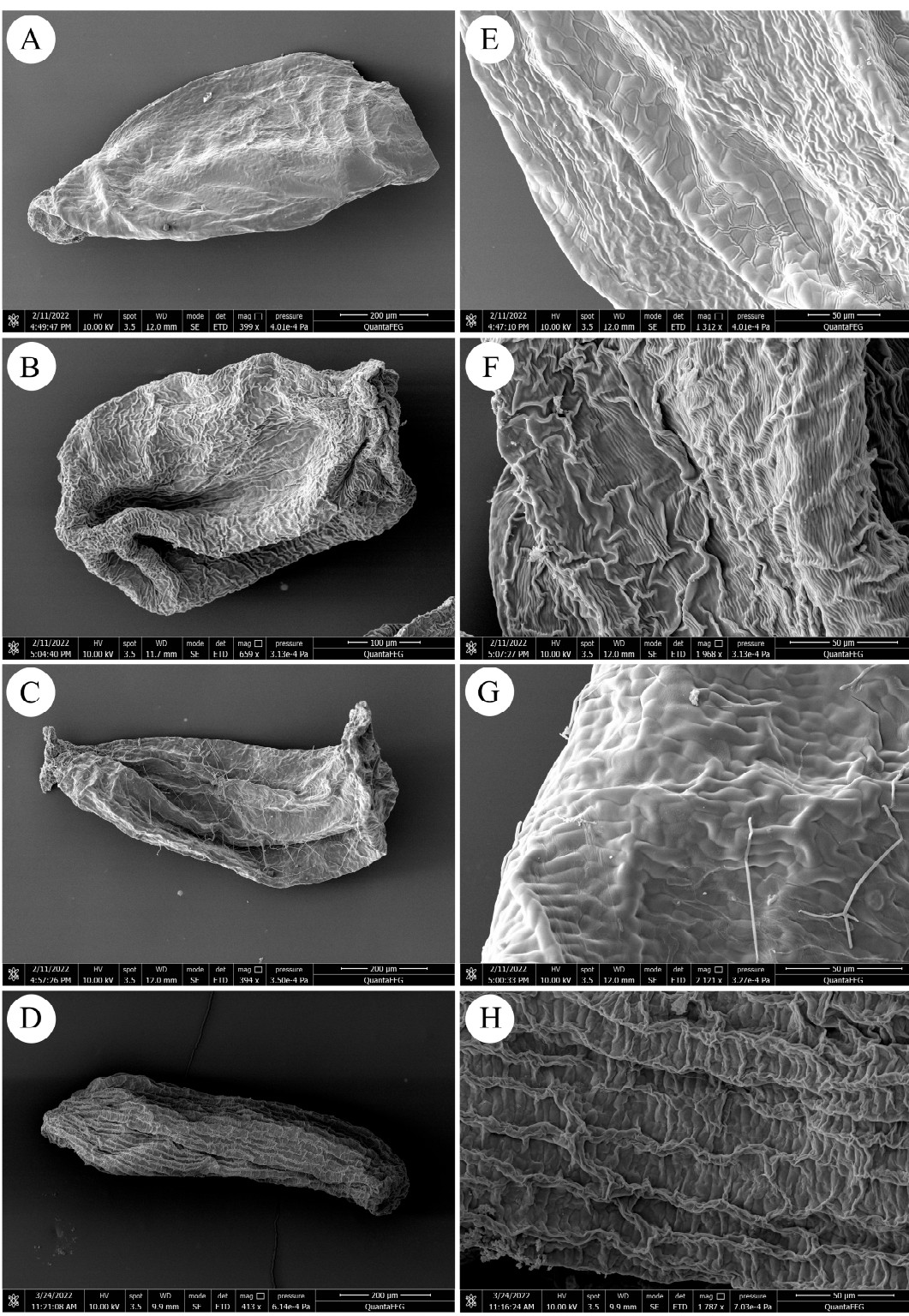

**Figure 2.** Scanning electron microscopy of four *Rhodiola* seeds. (**A**,**E**) *R. stapfii* (PEY0067558); (**B**,**F**) Lhünzê population (PEY0067593); (**C**,**G**) *R. prainii* (PEY0068682); (**D**,**H**) Namling population (PEY0067596); (**A**–**D**) The shape of seeds; (**E**–**H**) Local magnification of the seed micromorphology.

*3.4. Molecular Analyses*

Following the alignment, we obtained a matrix of 588 base pairs (bp) and selected SYM+G4 for the Bayesian and ML analyses (Figure S2). The 50% majority rule consensus tree of all the post burn-in trees is shown in Figure 3 with the Bayesian posterior probabilities (PPs).

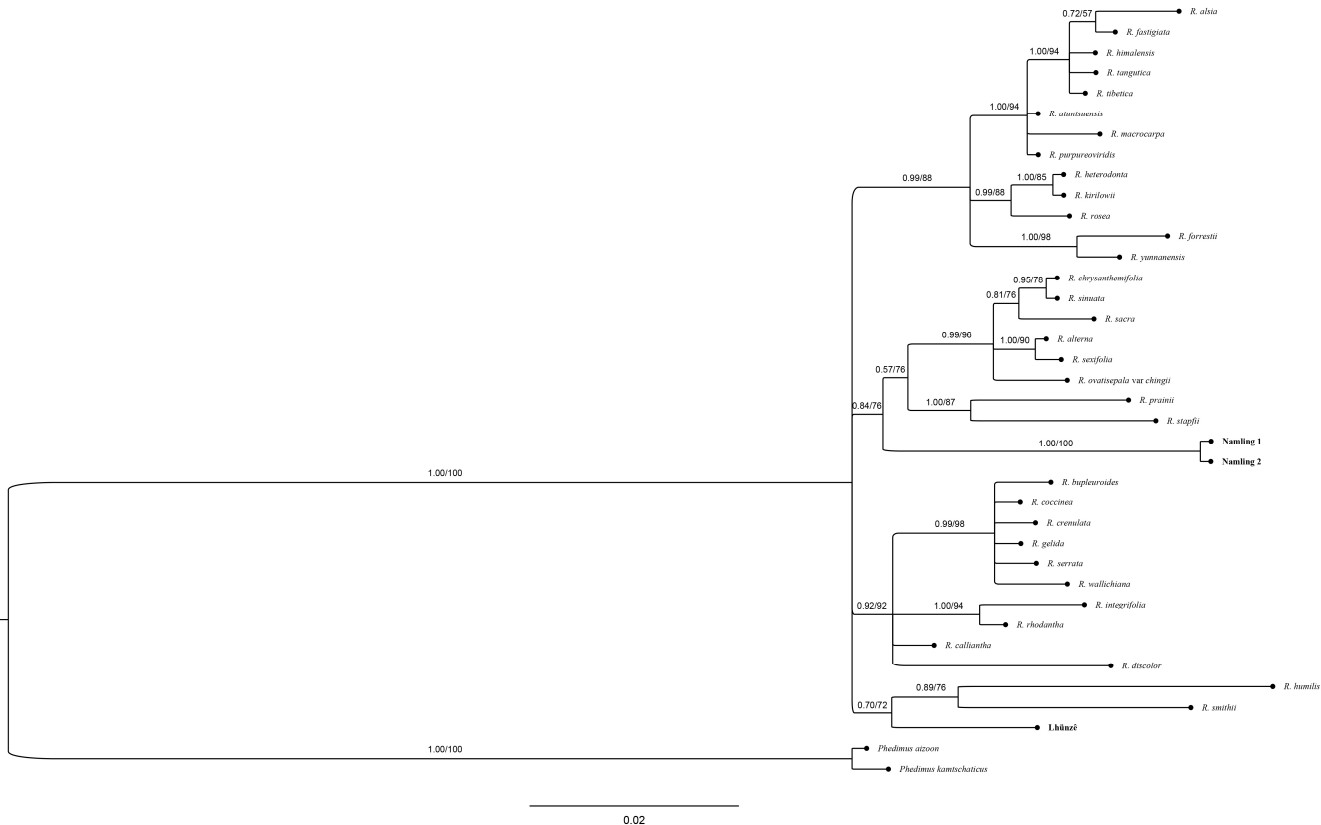

**Figure 3.** A Bayesian phylogenetic tree based on ITS sequences for the two new *Rhodiola* species described and their close relatives. The topology of the maximum likelihood (ML) tree was highly compatible with that of the Bayesian tree. Bayesian posterior probabilities (PPs: left) and bootstrap support (BP: right) values (>50%) are shown above the branch.

The BI tree strongly supported the monophyly of Sect. *Trifida*, Sect. *Prainia*, and Sect. *Pseudorhodiola*. The two samples of Namling populations from different sites are shown as a distinct clade (Posterior Probability [PP] = 1.0, Bootstrap value [BP] = 100%). The Namling populations formed a monophyletic clade with the species of Sect. *Trifida* and Sect. *Prainia* (PP = 0.84, BP = 64) (Figure 3), whereas the Lhünzê population formed a monophyletic clade with *R. smithii* and *R. humilis* (PP = 0.70, BP = 62).

## 4. Discussion

This survey found that species of *Rhodiola* have spread to the dry soil slopes of alpine meadow and beds of pebble beaches on the Qinghai–Tibet Plateau. It shows that *Rhodiola* has occupied various habitats on the Qinghai–Tibet Plateau. The habitat reflects the radiation of *Rhodiola*, which is consistent with the results of molecular systematics [15,25].

Seed micromorphology provides many characters that are potentially useful to identify species and perform phylogenetic inference in the Crassulaceae [26–28]. However, there has been little research on the seed coat micromorphology of *Rhodiola* [18]. In this study, we conducted an SEM analysis of the seed coats of *R. prainii, R. stapfii,* and the Namling and Lhünzê populations. The results show that the two morphological types of these four species differed from those previously reported [18]. The surface ornamentation of the seed coats of the Namling populations clearly differ from those of *R. prainii,* as do those of the Lhünzê population and *R. stapfii.* Thus, the seed coat characters of *Rhodiola* show

considerable diversity and have some value in the delimitation of species. Thus, there should be more focus on the seed coat micromorphologies of *Rhodiola*, which includes more than 70 species. Recently, many molecular systematic studies have been conducted, and *Rhodiola* has been an ideal material on which conduct research on radiated taxa [15,16,25,29], but few studies have integrated the ornamentation of seed coats.

In this research, we integrated morphological studies, SEM and ITS using PCA and a phylogenetic analysis for the delimitation of *Rhodiola*. A morphological comparison indicated that the Lhünzê population is similar to that of *R. stapfii*. In contrast, a phylogenetic analysis showed that the Lhünzê population is more closely related to *R. smithii* and *R. humilis* than *R. stapfii* phylogenetically. However, scale-like leaves were identified in the rhizome apex of the Lhünzê population, whereas the leaves in the rhizome apex of *R. smithii* and *R. humilis* were developed. The Namling populations are morphologically similar to those of *R. prainii*, but the PCA results show that they can be easily differentiated by the leaf shape (the length of the leaf, width of the leaf, length of between the widest part of blade and base of petiole and length of the petiole). A phylogenetic analysis shows that the Namling populations form a monophyletic clade with the species of Sect. *Trifida* and Sect. *Prainia*. In this monophyletic clade, the Namling populations were placed at the base.

Thus, based on the phylogenetic analysis, leaf morphological analysis and habitat information, we believe that the Lhünzê population and Namling population are distinct new species of *Rhodiola* that have not yet been described, and will provide detailed information to help understand the radiation of *Rhodiola*.

## 5. Description of the New Species

1. *Rhodiola wangii* S.Y. Meng sp. nov. (Figures 4 and 5)
　　urn:lsid:ipni.org:names: 77260717-1.

**Type:**—China. Xizang, Lhünzê Co., 28°06′00″ N, 91°55′53″ E, 4914 m, 20 July 2014, S.Y. Meng et al. MHW0071 (holotype PEY0067593, isotype, PEY0068681).

**Diagnosis:**—Similar to *Rhodiola stapfii* (Raymond-Hamet) S.H. Fu but differs in having an intensely broad rhombus and alternate leaves, a distinct petiole, stamens that are gathered together and have reflexed purple scales. (vs. middle stem leaves 5- or 6-verticillate, ovate to ovate-oblong, white nectar scales).

**Description:**—A low and delicate herbaceous perennial, 0.5–1 cm high. Caudex nearly erect, fewer branches, slightly thicker, usually 4–8 mm across; apical part often short, branched and accrescent, crowned by the scaly radical leaves. Scaly radical leaves membranous, persistent, purple, long ovate with an entire margin, 1.4–1.7 mm long, 0.8 mm wide. Roots very slender. Flowering stems 1–3 from the apex of each caudex branch, deciduous, 0.5–1 cm long, erect, simple, terete, smooth. Leaves alternate, flattened, broad rhombus, closely arranged on the stem, ascending-spreading, distinct petiole, spurless, glabrous, 5–7 mm long, 2–5 mm wide, entire. Inflorescences terminal, 1–3 flower buds form cymes, inconspicuous bracteates, bracts leafy. Flower green, 4(5)-merous, dioecious. Female flowers (♀) white-green, sepal 4 (5), free, long triangular, apex acuminate, entire, green, slightly fleshy, 1.2–1.6 mm long, 0.7–0.8 mm wide. Petals 4 (5), triangular free, green, with a short tip at the apex, 2 mm long, 0.8 mm wide, entire along the margin. Nectar-scales purple, trapezoid, rolling outward, truncate at the apex, 0.5 mm long, 0.4 mm wide. Carpel 4–5, free, erect, triangular ovate, ca. 2 mm long, style 0.5 mm long, curved outward. Follicles erect. Ovules ca. 5–9 in each locule. Seed orbicular-ovate ca. 0.8 mm long, 0.6 mm wide. Male flowers (♂) 4(5)-merous, sepals 4(5), green, free, long triangular, with a short tip at the apex, 1.2–1.5 mm long, 0.8–1.0 mm wide. Petals 4 (5) free, pale green, 1.8–2.0 mm long, 0.8–1.0 mm wide. Stamens 8–10, shorter than petals, antepetalous ones inserted 0.5 mm from petal base, long ca. 1.8 mm, antesepalous ones ca. 2 mm long, anther yellow. Nectar scales 4(5), broadly quadrangular, purple-red, 0.5 mm long, 0.2 mm wide. Carpel 4(5), erect, undeveloped.

**Distribution and Habitat:**—Perennial herb on mountain slopes, 4914 m. The distribution of *R. wangii* S.Y. Meng is somewhat limited (Figure 6). To date, only one population has been found on an arid hillside in the southeast region of Xizang, China.

**Phenology:**—Flowers from July to August, and fruits from August to September.

**Etymology:**—The epithet 'wangii' is used to commemorate the famous Chinese botanist and plant science popularization pioneer, Professor Wang Jinwu, Peking University. He published many popular articles and an illustrated handbook of botanical taxonomy, which promoted public attention and love for plants.

**Common name (assigned here):**—Jing Wu Hong Jing Tian (劲武红景天; Chinese name).

**Proposed IUCN conservation status:**—The new species grows in the arid meadows of Mt. Shangala. We collected only one population near Mt. Shangala, while another population was found between the road of Lhünzê County to Comai County in Lhünzê County in 2021. Although the habitat of the Qinghai–Tibet Plateau is relatively stable, more active economic and construction activities, such as grazing, may affect the population. The species is considered to be "Vulnerable" (VUD1) according to the IUCN Red List Criteria [30].

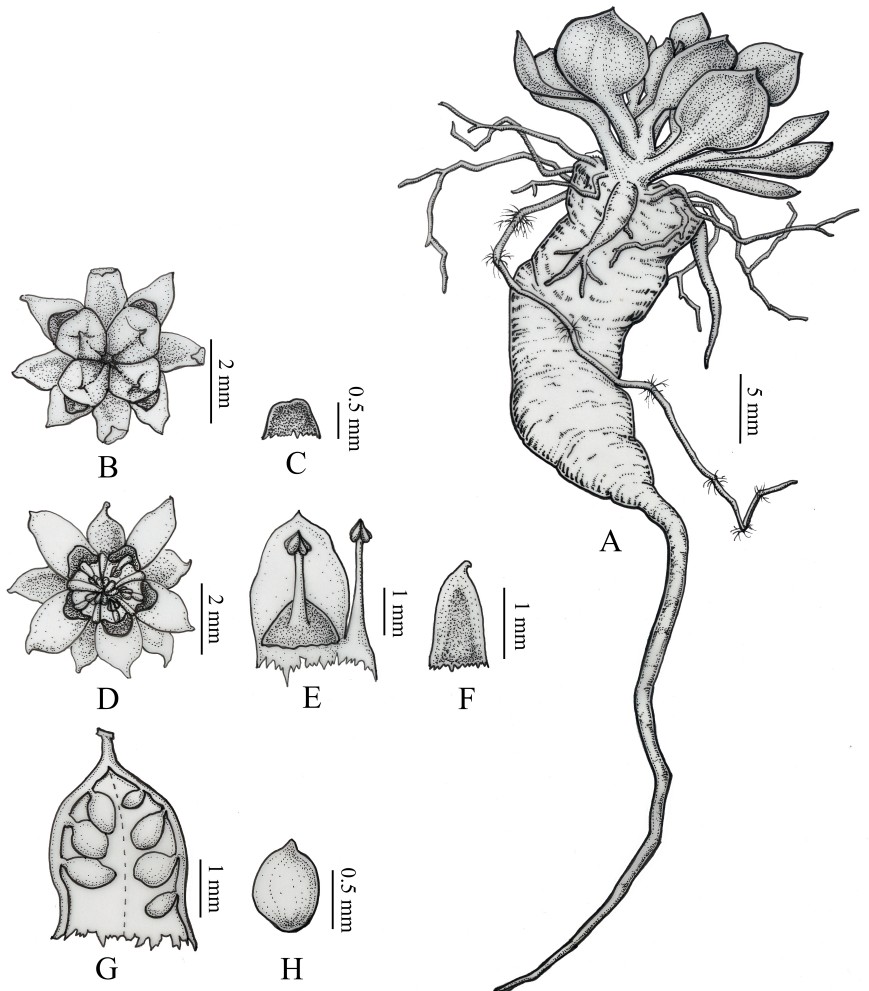

**Figure 4.** *Rhodiola wangii* S.Y. Meng. (**A**). Rhizome and flowering stem; (**B**). Female flower; (**C**). Nectar; (**D**). Male flower; (**E**). Petal and Stamens; (**F**). Sepal; (**G**). Follicles; (**H**). Seeds. Drawing by Ye Lv from PEY0067593.

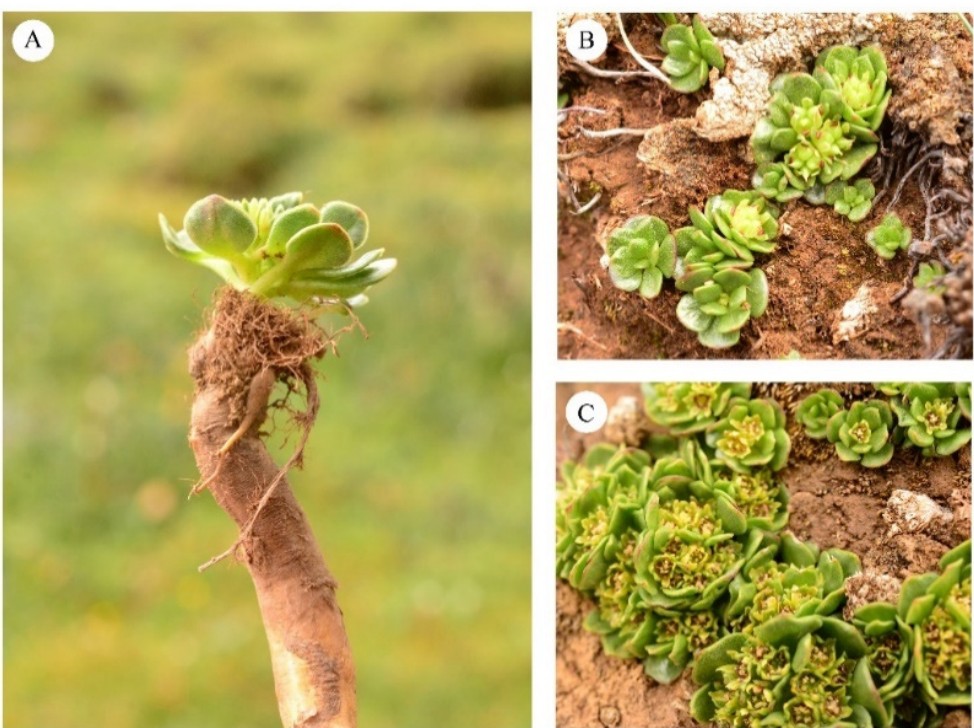

**Figure 5.** *Rhodiola wangii* S.Y. Meng. (**A**). Rhizome and flowering stem; (**B**). Female plants; (**C**). Male plants.

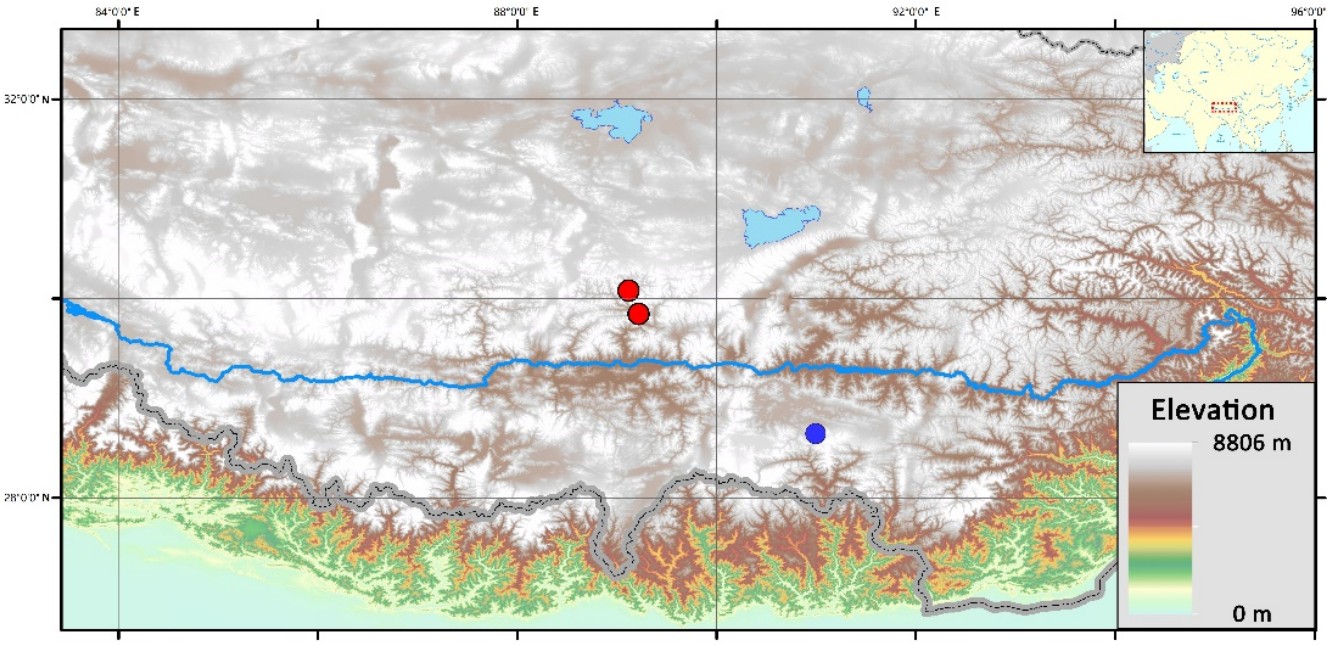

**Figure 6.** Geographic distribution. Blue dot shows *Rhodiola wangii* S.Y. Meng. Red dots show *R. namlingensis* S.Y. Meng.

2. *Rhodiola namlingensis* S.Y. Meng sp. nov. (Figures 7 and 8)
urn:lsid:ipni.org:names: 77260718-1.

**Type:**—China. Xizang: Namling Co., 30°04′91″ N, 89°06′27″ E, 4208 m, 18 September 2015, S.Y. Meng et al. 2015091802 (holotype, PEY0067596; isotype PEY0067595).

**Diagnosis:**—Similar to *Rhodiola prainii* (Hamet) H. Ohba but differs in its exerted alternate leaves, more than four stem leaves, thick leaf blades, obovate to inverted triangle, short petiole (vs. stem leaves 4, verticillate, oblong-elliptic leaf blades, ovate, broadly ovate, or reniform-orbicular, base abruptly narrowed to long attenuate.)

**Description:**—A perennial herb, 2–4 cm tall. Roots slightly thicker, branches, 4–9 cm long. Caudex cylindrical, slender, 2–7 mm thick, erect, the apical part densely covered with scaly radical-leaves. Scaly radical-leaves broadly triangular with entire margin, persistent, acuminate at the apex, 1.5–1.8 mm long, 1.5 mm wide, reddish brown. Flowering stems 1–4 from each branch apex of rhizomes, erect, simple, terete, glabrous. Leaves alternate, densely arranged on the upper part, ascending-spreading, spurless, thick herbaceous, flattish, yellowish green, obovate to inverted triangle, round at the apex; very short attenuate at the base; entire along the margin, 12–16 mm long, 6–8 mm wide, glabrous on both surfaces. The costa not obvious, with a short petiole. Inflorescences simple or few branches, corymbiform, 1–4 flowered; bracts shortly petiolate, obovate, 5–7 × 3–4 mm. Flowers bisexual, white or red, unequally 5-merous; pedicel 4–6 mm. Sepals 5, green, succulent, triangular-ovate, 5–7 mm long, 3–3.5mm wide, obtuse at the apex, entire. Petals 5, white or red, sometimes white with red plaques, oblong-ovate, 10–12 mm long, the united part 0.5 mm long, the free part 9–11 mm long, 6–8 mm wide, apex acuminate, tapering to the base. The petals do not fully open at maturity but stand upright surrounded by stamens and ovaries. Stamens 10, slightly shorter than petals, antepetalous ones inserted 2–3 mm from petal base, long ca. 5 mm, antesepalous ones 7–8 mm long, the filaments slender, sub-linear, anther tip, blue. Nectar scales trapezoid, ca. 1 mm long, ca. 0.8 mm wide, apex emarginate. Carpels 5, slightly connate at base, erect, long elliptic, 6–7 mm long, style 2–3 mm long, erect; each carpel has about 20 ovules. Seeds long elliptic, 1.5 mm long and 0.5 mm wide.

**Distribution and Habitat:** *R. namlingensis* was only known from southeast Xizang (Namling), China (Figure 6). Now, two populations have been found. It grows in gravel crevices or rock crevices on the beach of the Jiacuo Zangbo Valley, between 3800 and 4000 m above sea level.

**Etymology:**—The specific epithet is derived from Namling county, southeast of Xizang, China.

**Phenology:**—Flowering in August to September, fruiting in September to October.

**Common name (assigned here):**—Nan Mu Lin Hong Jing Tian (南木林红景天; Chinese name).

**Proposed IUCN conservation status:**—This new species is only known from southeast Xizang (Namling) where two populations were found in gravel crevices or rock crevices on the beach of the Jiacuo Zangbo Valley, between 3800 and 4000 m above sea level. The species is considered to be "Vulnerable" (VUD1) according to the IUCN Red List Criteria [30].

**Additional specimens examined (paratype):**—China. Xizang: Namling Co., 4313 m, 18 September 2015, S.Y. Meng et al. 2015091804 (PEY0068680, PEY0067594); Namling Co., 4313 m, 30 July 2015, S.Y. Meng & Z.M. Wang MWH110 (PEY0066822); Namling Co., 4212 m, 11 August 2010, S.Y. Meng msy05 (PEY0068679).

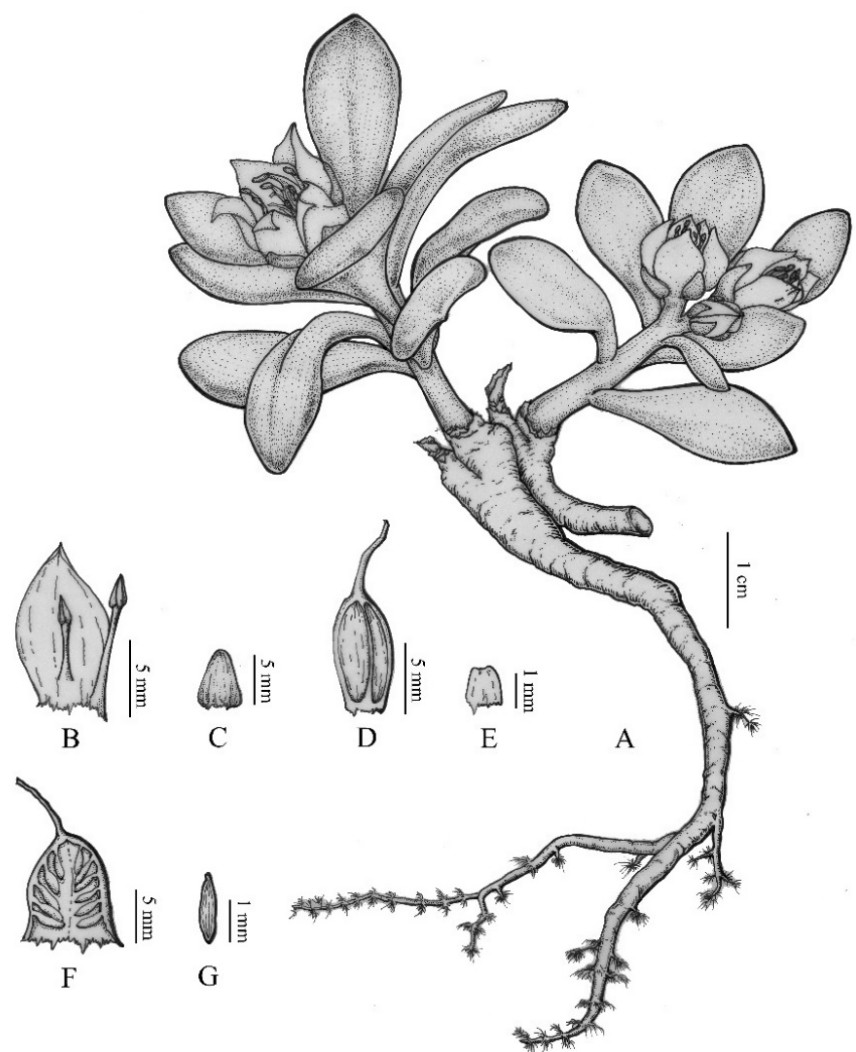

**Figure 7.** *Rhodiola namlingensis* S.Y. Meng. (**A**). Rhizome and flowering stem; (**B**). Petal and stamens; (**C**). Sepal; (**D**). Carpel; (**E**). Nectar; (**F**). Follicle; (**G**). Seed. Drawing by Ye Lv from PEY0067595.

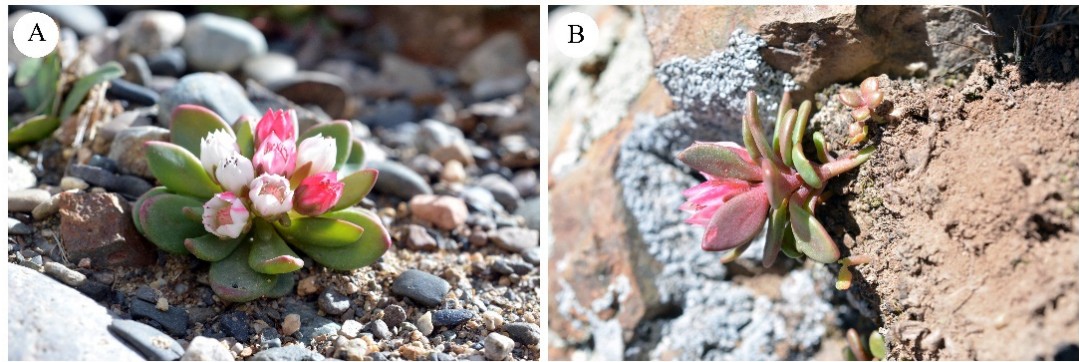

**Figure 8.** *Rhodiola namlingensis* S.Y. Meng. (**A**). A plant and its habitat; (**B**). Side face of the plant.

**Supplementary Materials:** The following supporting information can be downloaded at: https://www.mdpi.com/article/10.3390/d14040289/s1, Figure S1: Leaf measurement. Figure S2: ML tree based on ITS sequences for these two new *Rhodiola* species and their close relatives. Table S1: Plant materials of 36 accessions of the Rhodiola taxa and two outgroup taxa with their collection locality, voucher information, and accession numbers of ITS sequences reported by Zhang et al. (2014).

**Author Contributions:** The experimental design was completed by S.M. and Z.W. Sample collection and treatment were conducted by S.M. The data were analyzed, and the figures and tables prepared with assistance from L.Y. and Z.W. The manuscript was drafted by S.M. edited the manuscript for structure, language, and scientific content. All authors have read and agreed to the published version of the manuscript.

**Funding:** This research was funded by the National Natural Science Foundation of China (No. 31600159).

**Institutional Review Board Statement:** Not applicable.

**Data Availability Statement:** Not applicable.

**Acknowledgments:** The authors thank the Core Facilities at the School of Life Sciences, Peking University for assistance with scanning electron microscope observation. We also thank Hong-ya Gu, Guang-yuan Rao and Sodmergen, Jia-chen Hao, and Yi-hao Shi for field assistance.

**Conflicts of Interest:** The authors declare no conflict of interest.

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
