# Peer review of "Integrative Taxonomy Supports Two New Species of Rhodiola (Crassulaceae) in Xizang, China"

_diversity, doi:10.3390/d14040289_

Round 1

Reviewer 1 Report

I found the manuscript on two new Rhodiola taxa from the Tibetan Plateau interesting and worth to be published on Diversity.

I just recommend authors to improve the Introduction section writing a bit more on the hypotheses of the speciation of this genus: it seems an ongoing case of differentiation maybe started after the LGM and still in progress, which leads new micro endemic taxa to appear in the area. At the same time, all the non appropriate sentences should be deleted.

The same is true in the Results section,  where the Authors include some ecological features of the newly described taxa in the morphological characteristics paragraph. 

If the Authors make these minor revisions to their manuscript, and some language revision, the new versionof the manuscript they will produce will be certainly approved for publication

Author Response

Dear Reviewers #1,

Reviewer 2 Report

Some expressions are less generally understood.

Fuchsia is expression for colour? Few errors in font: R. humilis.

Author Response

Dear reviewers #2,

Thank you for your letter and the reviewers’ comments concerning our manuscript entitled “Integrative Taxonomy Supports Two New Species of Rhodiola (Crassulaceae) in Xizang, China” (ID: diversity-1630616). Those comments are valuable and very helpful. We carefully read the comments and have made corrections. Based on the instructions provided in your letter, we uploaded the file of the revised manuscript. Revisions in the text are shown using red highlight for additions, and strikethrough font for deletions. The responses to the reviewer's comments are marked in red and presented following.

We greatly thank you for allowing us to resubmit a revised copy of the manuscript, and we highly appreciate your time and consideration.

All the best,

Shi-yong Meng

Reviewer 3 Report

The manuscript presented to me for review is a very important addition to the taxonomy of an interesting group of plants in an unexplored area of the world, but at this stage it cannot be published. Many, even basic mistakes and shortcomings lead me to reject it.

Below are only the most important objections, in pdf file more with suggestions and editorial comments. However, I see the potential in the most important content, this is why I proposes to the authors to seriously consider the construction of the manuscript and to refine it (especially the Introduction and Discussion) and to consider re-submitting a revised version of the manuscript.

The abstract and keywords need to be supplemented because they do not fully present the reviewed manuscript.

The introduction is very short, in fact it is one paragraph, the next of this chapter is not an introduction but a description of the research area,this chapter should be described in more detail; eg. genus of should be discussed more broadly, range of species etc ... There may be a description of the area but let it be the first, let the introduction begin with something general and go into detail.

The Introduction does not end with the clear aim of the research

The whole manuscript is quite a mess, the authors chose some features for research, without explaining their choice, later in the results there is no reference to these features, this and similar examples give the impression that the manuscripts are written sloppy. When we choosing some features for research, it is necessary to describe them in detail, their range of variability, etc.

In the manuscript we can find a few factual errors, the SEM image is not a morphology, as the authors write (line 149), but a micromorphology.

A serious mistake is the lack of reference to the herbarium numbers of individual tested specimens, including types! This in taxonomic manuscripts is unacceptable!

It is nowhere explained in the manuscript why the authors only study ITS and not also chloroplast DNA? Why such a choice?

Another example of chaos in this manuscript is the relation of what is in the results to the rest of the manuscript. In „Materials and methods”, the authors write about "morphological and SEM analysis", the next sub-chapter is "phylogenetic analysis" - while in the results in the sub-chapter "morphological analysis" there is a similarity between populations, mainly habitat with several of the features mentioned! After all, there should be only facts about the ranges of variation and all the characteristics listed in the methods! Also included is a SEM analysis which is not "morphological" but micromorphological.

Next, what is even more interesting, the authors compare other populations but without SEM, why? The SEM analysis itself gives very little data, in fact it is one sentence, because the shape of the seeds to be seen under a light-microscope, can it really not be read more? I think so?

PCA in Methods and Results should be described separately as „Statistical analyzes” resulting from „morphological studies”.

Besides, there is nothing at this manuscripy  why did you choose PCA (and not another analysis). But most importantly if PCA, what this analysis gave; which features discriminate best against the studied populations, which the least; no% explanation of the variability of both axes) – these are let's face it frankly very basic shortcomings, missing data that show a lack of rethinking of the written script.

The condition of the figures is a big objection as they are illegible, except for those showing new species.

The discussion is unacceptable, by the nature of things this chapter discusses the results with the literature in this manuscript - there is no reference to literature in this chapter. This is quite surprising and unacceptable in scientific work.

The manuscript is sloppy, a lot of editorial errors, the lack of italicized Latin names, repetitions, linguistic errors, defective entry of geographical coordinates I have indicated in the text. The authors must pay more attention to this and correct all shortcomings and gaps.

More comments in pdf file, I hope the authors will think carefully and improve this manuscript because it has potential, but not in this edition.

All the best

Author Response

Dear reviewers #3,

Thank you for your letter and the reviewers’ comments concerning our manuscript entitled “Integrative Taxonomy Supports Two New Species of Rhodiola (Crassulaceae) in Xizang, China” (ID: diversity-1630616). Those comments are valuable and very helpful. We read the comments carefully and made corrections based on them. Based on the instructions provided in your letter, we uploaded the file of the revised manuscript. Revisions in the text are shown using red highlight for additions, and strikethrough font for deletions. The responses to the reviewer's comments are marked in red and presented following.

All the best,

Shi-yong Meng

Round 2

Reviewer 3 Report

I appreciate the corrections made to the present appearance of the manuscript. The extended introduction, new data added to each of the chapters and subsections make the manuscript significantly more valuable and readable.

Repetitions are still disturbing, but I leave this matter to the authors for consideration. It's their manuscript, after all.

I only send minor corrections in the attached file, which in my opinion still need to be corrected, but at this stage I think that it can be easily published after selected corrections.

All the best 

GJW

Author Response

Dear Editors and Reviewers,

Thank you for your letter and for the comments of reviewer concerning our manuscript entitled “Integrative Taxonomy Supports Two New Species of Rhodiola (Crassulaceae) in Xizang, China” (ID: diversity-1630616). Those comments are valuable and very helpful for improving our paper. We revised all the comments carefully and hope meet with approval. Revised portion are marked in red in the paper.

Thank you very much for your consideration.